# Development and Characterization of Interstitial-Fluid-Mimicking Solutions for Pre-Clinical Assessment of Hypoxia

**DOI:** 10.3390/diagnostics13193125

**Published:** 2023-10-04

**Authors:** Nadia Muhammad Hussain, Bilal Amin, Martin O’Halloran, Adnan Elahi

**Affiliations:** 1Translational Medical Device Lab, University of Galway, H91 TK33 Galway, Ireland; bilal.amin@universityofgalway.ie (B.A.); martin.ohalloran@universityofgalway.ie (M.O.); adnan.elahi@universityofgalway.ie (A.E.); 2Electrical and Electronic Engineering, University of Galway, H91 TK33 Galway, Ireland; 3School of Medicine, University of Galway, H91 TK33 Galway, Ireland

**Keywords:** birth asphyxia, buffer, electrical conductivity, hypoxia, impedance, interstitial fluid, metabolic acidosis, metabolic alkalosis, pH

## Abstract

Asphyxia, a leading cause of illness and death in newborns, can be improved by early detection and management. Arterial blood gas (ABG) analysis is commonly used to diagnose and manage asphyxia, but it is invasive and carries risks. Dermal interstitial fluid (ISF) is an alternative physiological fluid that can provide valuable information about a person’s health. ISF is more sensitive to severe hypoxia and metabolic disorders compared to blood, making it an attractive option for minimally invasive asphyxia detection using biosensors. However, obtaining ISF samples from humans is challenging due to ethical concerns and sampling difficulties. To address this, researchers are developing ISF-mimicking solutions as substitutes for early testing and evaluation of biosensors. This paper focuses on the development of these solutions for bench-based testing and validation of continuous asphyxia-monitoring biosensors. With an understanding of the factors influencing system quality and performance, these solutions can aid in the design of biosensors for in vivo monitoring of dermal ISF. Monitoring interstitial fluid pH levels can provide valuable insights into the severity and progression of asphyxia, aiding in accurate diagnosis and informed treatment decisions. In this study, buffer solutions were prepared to mimic the pH of ISF, and their electrical properties were analyzed. The results suggest that certain buffers can effectively mimic metabolic acidosis associated with asphyxia (pH < 7.30), while others can mimic metabolic alkalosis (pH > 7.45). Overall, this research contributes to the development of ISF-mimicking solutions and lays the groundwork for biosensor systems that monitor dermal ISF in real time.

## 1. Introduction

Hypoxia is a condition that occurs when there is a reduced oxygen supply to the body’s tissues, while birth asphyxia is oxygen deprivation that happens around the time of birth due to various perinatal events [1]. Hypoxia can result in a spectrum of acid–base changes and physiological disorders at the cellular, tissue, or organ level [2]. The global annual burden of this disease is about four million cases, leading to one million neonatal deaths [3]. Birth asphyxia and subsequent metabolic acidosis (neonatal acidemia) are also associated with early neonatal death, which is defined as the death of a newborn between zero and seven days after birth, and it accounts for 73% of all postnatal deaths worldwide. Additionally, birth asphyxia can lead to multiorgan dysfunction, hypoxic–ischemic encephalopathy, seizures, cerebral palsy, long-term neurological sequelae, and neonatal mortality [4].

Arterial blood gas (ABG) analysis is considered essential for the accurate diagnosis and clinical management of birth asphyxia [5]. ABG is a diagnostic tool for measuring oxygen, carbon dioxide, and acid–base disorder in the blood [5]. In ABG analysis, blood is the primary source for identifying biomarkers for the accurate diagnosis of asphyxia [6]. Neonatal blood sampling requires skilled handling to ensure sufficient sample acquisition while also minimizing the risk of complications such as bleeding, infection, and vascular damage due to the small size and fragility of blood vessels. Additionally, the limited volume of blood that can be safely obtained from neonates increases the difficulty of sampling, necessitating careful selection of appropriate tests and techniques to avoid unnecessary patient discomfort or harm [7,8]. Despite ABG’s potential clinical advantages, the non-continuous mode and discomfort associated with invasive blood sampling are the key drawbacks of ABG [6]. As an alternative to blood, interstitial fluid (ISF) is a more sensitive physiological indicator of asphyxia. Mild metabolic disorders can be effectively buffered by blood, which helps maintain arterial blood pH within the normal range [9]. However, due to the limited buffering capacity of ISF, pH levels quickly change in response to hypoxic events, making ISF more susceptible to even small changes in pH levels resulting from severe hypoxia. This can cause significant deviations from the normal pH range in the ISF [9]. In the last two decades, ISF has been used for minimally invasive detection of various health conditions such as hereditary metabolic disorders, organ failure, and therapeutic efficacy [9]. Additionally, ISF’s composition is similar to that of blood due to the equilibrium maintained by small molecules, such as carbon dioxide, phosphates, and albumin. This similarity in composition makes ISF an interesting alternative to blood analysis for clinical applications, as it allows for non-invasive sampling while maintaining the same physiological outputs [10]. Due to ISF’s advantages in terms of volume and composition, ISF is a promising alternative for continuous sampling and monitoring of hypoxia-associated metabolic disorders and asphyxia [11].

Various techniques have been developed for the extraction of ISF from the skin, including suction blisters, open-flow microperfusion, and microdialysis [12]. However, these techniques can be complicated, require trained professionals, and use bulky instruments that may cause discomfort to patients [13]. In contrast, microneedle patches are a minimally invasive, continuous, rapid, and cost-effective technology that overcomes the challenges associated with dermal ISF sampling [14]. The reduced invasiveness of ISF sampling and the sensitivity of ISF to severe hypoxia (asphyxia) and metabolic disorders compared to blood makes ISF an attractive target for biosensor-based detection of asphyxia in a minimally invasive manner [11]. To develop these sensors, researchers require physiologically representative liquid samples that mimic the properties of ISF. This need is particularly critical for the development of a universal biosensor for continuous and minimally invasive asphyxia monitoring. ISF-mimicking solutions could provide a simple and rapid solution for the pre-clinical testing and validation of continuous asphyxia-monitoring biosensors.

pH, as a quantitative measure of solution acidity or alkalinity, plays a crucial role in predicting asphyxia. Low pH levels (acidosis (pH < 7.30)) reliably indicate the occurrence of asphyxia, whereas high pH levels signify the occurrence of metabolic alkalosis (pH > 7.45) [15], potentially arising from inadequate management of oxygen supply during asphyxia interventions [16]. Monitoring pH levels in bodily fluids such as ISF can provide healthcare professionals with valuable insights into the severity and progression of asphyxia, facilitating accurate diagnosis and treatment decisions [17]. The pH of body fluids has been thoroughly investigated, and an average value of pH 7.4 is generally reported [18]. However, the pH of ISF and its variability have not been well investigated in the literature. The majority of published evidence indicates that under healthy settings, the pH of ISF is similar to the pH of blood, i.e., 7.35–7.45 [18,19]. ISF-mimicking solutions could support the design of initial systems of biosensors for in vivo dermal ISF by enabling an understanding of the variables influencing the system’s quality and performance [20].

The objective of this study was to develop ISF-mimicking solutions with the same pH as normal ISF, as well as hypoxic ISF characterized by acidosis or alkalosis. In addition, this study aimed to investigate the relationship between the pH of ISF and its electrical properties, including impedance and conductivity, under normal and hypoxic conditions. This work allows for the subsequent development and testing of an impedance-based biosensor for the detection of hypoxia-induced pH changes in ISF. A series of buffer solutions were prepared with varying pH ranges, and their electrical properties were characterized by measuring their impedance using a four-probe method. The impedance data were then converted to conductivity data, and eight mimicking solutions were selected based on their electrical conductivity values falling within the required conductivity range (0.41–0.80 S/m) of ISF. The experimental procedure began with the measurements of the pH and impedance of the buffers at their actual pH values to observe acidosis without any adjustments made to the pH levels. Subsequently, hypoxic acidosis was induced by adding hydrochloric acid (HCl). Following this, alkalosis was observed without adjusting the pH values of the buffers. Finally, the addition of sodium hydroxide (NaOH) was used to observe the effect of alkalosis in the buffering solution. The process of selecting the most appropriate buffers to mimic ISF involved evaluating their pH values, pH–impedance plots, and conductivity across a frequency range of 10 Hz to 100 kHz. This selection was made by comparing the properties of various buffers and choosing the ones that exhibited the closest resemblance to ISF.

## 2. Materials and Methods

### 2.1. Selection of Mimicking Materials

In this study, a range of buffers were prepared with pH values ranging from 6 to 8. This range of pH mimics the impact of severe-hypoxia-induced acidosis and poor-hypoxia-management-induced alkalosis. The buffer systems and their corresponding pH ranges are tabulated in Table 1.

The selection of a buffer for the preparation of ISF-mimicking solutions is primarily determined by the pH value of the buffer, which serves as the fundamental characteristic defining its capacity to maintain a stable pH environment [15]. Other key parameters include the buffer capacity or pKa value, the solubility of the buffer in water, temperature, and the concentration and ionic strength of its components [19]. Optimal buffering is achieved when the pH value of the buffer system is within one unit of its pKa value, which can be expressed mathematically by Equation (1) [21]:pH = pKa ± 1(1)

In addition to pH, other factors including the stability and nature of the buffer are also considered in selecting appropriate buffers. For instance, although imidazole-HCl buffer has a pH range of 6.2–7.8, it tends to bind with various metals and is largely unstable [22]. Moreover, when creating buffers in the laboratory, the hazardous nature of the buffer is also taken into account [23]. Cacodylate, for instance, is a hazardous buffer that, when combined with a low pH value, can lead to enzyme inactivation [23]. Despite these challenges, all prepared buffers and reagents are commercially available and have a relatively low cost.

A simple and effective method for determining whether an aqueous solution of a conjugate acid/base pair functions as a buffer is the use of the Henderson–Hasselbalch Equation (2) [24]:(2)$$pH=pKa+log(\frac{\left[A^{-}\right]}{\left[HA\right]})$$
where Ka is the acid disassociation constant, pKa is the negative logarithm of Ka, HA is the concentration of the acid, and A^−^ is the concentration of the conjugate base. This equation allows for the calculation of the acidity of the buffer solution by considering the pKa of the acid and the ratio of the concentrations of the conjugate acid and base forms of the buffer.

The pH of interstitial fluid (ISF) is a critical clinical biomarker for determining asphyxia in neonates and ISF phantom design due to its rapid response to metabolic disturbances, clinical relevance, and specificity for acidosis assessment [16,25]. Furthermore, pH is a primary factor influencing electrical conductivity, which is essential for developing and testing impedance-based sensors for asphyxia detection [26,27]. While carbon dioxide (CO_2_) and oxygen (O_2_) levels are also important in assessing neonatal asphyxia, pH is considered more critical because it provides a comprehensive view of the overall acid–base status of the infant. It reflects not only the degree of respiratory distress (as indicated by CO_2_ levels) but also the metabolic consequences of oxygen deprivation (as indicated by lactate accumulation) [15,16]. Therefore, in this study, pH is prioritized as the primary biomarker in the development of electrical conductivity/impedance-based ISF phantoms.

### 2.2. Preparation of Buffers

Henderson and Hasselbalch’s equation (Equation (2)) was used to make buffered solutions by determining the precise amounts of acid and base required to achieve the desired pH [24]. The preparation of a buffer involves several steps, including weighing the chemical reagents with high precision and accuracy of the weighing scale, dissolving them in de-ionized water, and measuring the resultant pH with the calibrated and accurate pH meter (Thermo Electron Corporation 420A+, Waltham, MA, USA).

The flow sheet diagram of the stepwise process of the preparation of the buffers is shown in Figure 1 and Figure 2. Because the acid-to-base ratio in a buffer is directly tied to the resulting pH, it is critical to utilize precision and accuracy when preparing a buffer by utilizing equipment such as a weighing scale, pipets, and a pH meter [28]. For accurately weighing the chemical reagents, an analytical balance (Mettler Toledo AB54, Columbus, OH, USA) with a readability of 0.1 mg and repeatability of 0.08 mg was used. Before measuring each chemical reagent, the weighing container or paper was cleared by placing the container on top of the balance, effectively canceling out its weight and establishing a zero-reference point. The pH meter was used to record the pH values of each buffer solution. The pH meter was calibrated with standard calibrated pH solutions (pH = 4.00, pH = 7.00, pH = 10.00). These standard buffer solutions with a tolerance of ±0.02 pH units were purchased from Fischer Scientific, Dublin, Ireland. The pH sensor was cleaned with de-ionized water and was dried before each measurement.

The concentrations of most buffers that operate optimally are between 0.1 M and 10 M. When preparing the buffer solution, to keep consistency throughout the experiment, the concentration of each buffer solution was kept constant, 0.1 M, and the total volume of the buffered solution was 0.1 L. HCl and NaOH were used to introduce the effects of metabolic acidosis and metabolic alkalosis, respectively.

The conductivity of all the buffers was calculated from the recorded impedance values. As the conductivity values are temperature-dependent, to provide a consistent point of reference for comparison, standard conditions were established at room temperature, which is typically defined as 22 ± 3 °C. All the measurements were completed at room temperature, and the temperature of each buffer was measured by using a digital thermometer with a resolution of 0.01 pH units and an accuracy of ±0.005 units.

If the buffers were required to be prepared by mixing the stock solution, the required mass of the reagents to prepare the stock solution was calculated by using Equations (3) and (4) [29].
(3)$$M_{w}=\frac{m}{n_{i}}$$
where $M_{w}$ = molar weight or molar weight (g/mol), m = mass of the substance in grams (g), and $n_{i}$ = number of moles of a substance.
(4)$$c=\frac{n}{V}$$
where c = molar concentration/molarity (mol/L), n = moles of the solute (mol), and V = final volume of the solution (L). The concentration of resulting solutions using stock solutions can be determined by using Equations (5) and (6) as follows:(5)$$C_{s}V_{s}=C_{d}V_{d}$$
where $C_{s}$ = stock solution’s concentration, $V_{s}$ = volume of stock solution, $C_{d}$ = dilute solution’s concentration, and $V_{d}$ = volume of the dilute solution. The buffers are prepared at different pH values and are described with their reagents in Table 2.
(6)$$Dilution\;Factor=\frac{C_{s}}{C_{d}}=\frac{V_{d}}{V_{s}}$$

### 2.3. Measurement Setup

Two types of measurements were performed on each buffer solution. The pH of each buffer solution was measured using a digital pH meter (Orion model 420Aplus, Thermo Fischer Scientific, Waltham, MA, USA) and a mercury-free pH combination electrode (Thermo Fischer Scientific, Waltham, MA, USA). Before the measurements, the pH meter was calibrated using standard buffers of pH 4.0, 7.0, and 10.0. After allowing the reading to stabilize for 30 s, the pH was recorded. The temperature was also recorded using a digital thermometer (Hanna Instruments, Smithfield, RI, USA, model 1204, Thermo Fischer Scientific, Waltham, MA, USA).

For impedance measurements, the impedance measurement system was set up as displayed in Figure 3. The measurements were recorded using a PGSTAT204 (Autolab, Kanaalweg Den Haag, The Netherlands), which was set up in galvanostatic mode at room temperature (23 °C). The readings were obtained using a personal computer with Nova 2.1 software and the FRAM32 Impedance Analysis module (Metrohm Autolab B.V., Utrecht, The Netherlands). The measurement setup consisted of the PGSTAT204 connected to a four-point collinear probe, which consisted of a working electrode, counter electrode, reference electrode, and sensing electrode arranged in a collinear pattern [30]. The galvanostatic mode technique involves the flow of a constant electric current of 100 µA between the inducing electrodes.

The four-electrode collinear probe was fabricated from pure copper with a thickness of 0.5 mm and had electrode diameters of 3 mm and inter-electrode spacings of 2 mm. The probe was characterized, as described in [31], using saline solutions of concentrations 0.01 M (M = molar = mol/L), 0.05 M, and 0.1 M over the frequency range of 10 Hz to 100 kHz [32]. The characterization of the impedance probe involves the estimation of the cell constant: The cell constant relates the measured conductance to the corresponding reference conductivity. The cell constant of the collinear probe was calculated using the Pearson correlation coefficient (R) between the measured conductance and reference conductivity of the saline solutions. The cell constant for the proposed collinear probe was determined to be 0.032 m within the frequency range from 10 Hz to 100 kHz.

The accuracy of the probe was assessed using a 0.01 M potassium chloride solution. The accuracy of the probe was evaluated by measuring the impedance within the frequency range of 10 Hz to 100 kHz. The cell constant and the measured conductance were used to calculate the conductivity of the potassium chloride solution [32]. The probe with the best accuracy with a maximum difference of 10% was selected to record impedance measurements. The results indicated that the designed probe delivered consistent, precise, and reliable impedance measurements.

To collect impedance readings, the frequency sweep was set up at 05 logarithmically spaced steps between 10 Hz and 100 kHz. The conductivity was calculated from the measured frequency-dependent complex impedance data and the cell constant using Equations (6)–(10) [20]. These equations explain the steps used to calculate conductivity.
(7)Z=R+jX
where Z = impedance in ohms (Ω), R = resistance (Ω), and *X* = reactance (Ω).
(8)Y=G+jB,
where *Y* = admittance in siemens (S), *G* = conductance (S), and *B* = susceptance (S).
(9)$$G=\frac{R}{R^{2}+X^{2}},$$
(10)$$\sigma =\frac{G}{k}$$
where *σ* is the conductivity of buffer in siemens per meter (S/m) and *k* is the cell constant (m).

The temperature and pH of each buffer solution were recorded before the impedance was measured. The impedance measurements were repeated three times, and the mean impedance values were taken to calculate the conductivity for each buffer. Additionally, before taking each measurement, the thermometer, pH electrode, and probe were cleaned with distilled water and dried in the air. The diagram of the measurement system is shown in Figure 3.

The operating frequency range of 10 Hz to 100 kHz was chosen based on prior research [32] that utilized impedance spectroscopy to examine blood impedance. Frequencies above 1 MHz have been shown to generate stray capacitances and inductances that can potentially compromise the accuracy of measurements [33]. According to Buendia et al. [34], the susceptance at high frequencies is exclusively attributed to parasitic capacitance. Thus, the slope of the graph at high frequencies is equivalent to the level of parasitic capacitance that is present in the experimental setup. For tissue conductivities, parasitic capacitances represent a significant source of systematic error that can result in a reduction in the measured impedance modulus at frequencies greater than 200 kHz [34].

The correlation between the measured pH values and the impedance of each buffer was plotted using MATLAB2020b software.

## 3. Results and Discussion

### 3.1. Impedance Measurement of Buffer Solutions

The ISF-mimicking buffer solutions were selected by comparing the pH value, conductivity, and impedance with the corresponding reference values of ISF as documented by Pockevicius et al. [35]. The correlation between the pH of ISF and ISF impedance has not been extensively studied. However, based on the inverse relationship between the pH and impedance of cerebrospinal fluid (CSF) as documented by Baumann et al. [36], it can be inferred that a similar inverse relationship exists between the pH and impedance of ISF. This means that as the pH of ISF decreases (acidosis), the impedance value of ISF is likely to increase and vice versa. This inference is made because ISF and cerebrospinal fluid (CSF) are highly comparable in their composition and function [37]. As components of the extracellular fluid in the body, both ISF and CSF serve to support homeostasis and provide protective cushioning for tissues [38]. These fluids exhibit similarities in their clear appearance and their respective compositions, including high sodium and chloride ion concentrations and low levels of protein, which are integral in maintaining normal physiological conditions [38,39]. While there are no impedance data for ISF available in the current literature, inferences about impedance trends can be made based on the expected changes in ionic conductivity due to pH changes. In the context of bioimpedance, changes in pH can impact the electrical impedance of biological tissues, as the electrical impedance of tissue is dependent on its ionic conductivity [40]. For example, if a tissue becomes more acidic (acidosis or asphyxia), its ionic conductivity may increase, leading to a decrease in its electrical impedance [40]. Similarly, if a tissue becomes more alkaline (alkalosis), its ionic conductivity may decrease, leading to an increase in its electrical impedance [40]. Therefore, the ISF conductivity value is considered a reference value to make ISF phantoms.

### 3.2. Selection of ISF-Mimicking Solutions Based on Impedance Measurements

Following the completion of buffer preparation, the goal in the preliminary stage was to identify the buffers that adhere to the inverse relationship between pH and impedance. Out of the eight buffers tested (as listed in Table 2), five (CBC, BES, SPB, PPB, and PCB) exhibit the expected inverse relationship between pH and impedance. This relationship is illustrated in Figure 4 for these five buffers. For example, Figure 4b illustrates the inverse relationship for the BES buffer between pH and impedance, where impedance decreases as pH increases from 6.6 to 7.43. The impedance values were measured at five different frequencies (10 Hz, 100 Hz, 1 kHz, 10 kHz, and 100 kHz) and are depicted by various colored lines on the graph. The red line represents the average frequency line. In this study, it was observed that as the pH value increases, the impedance values at various frequencies tend to approach the average impedance value. Therefore, the average impedance value was considered as the significant value for comparing the relationship between pH and impedance. Conversely, three buffers (CBTP, BTP, and Trizma) demonstrated a direct relationship between pH and impedance. This relationship is demonstrated in Figure 5, where an increase in pH results in a corresponding increase in impedance.

These three buffers have been identified to exhibit the alkalosis effect, and additional experiments were conducted to investigate whether the ISF could reproduce this effect while adhering to the reference ISF values.

Based on the limited existing research on inducing acidosis and alkalosis in ISF ex vivo experiments, the metabolic acidosis effect has been demonstrated using a previous study on rats, which demonstrated that the introduction of acidosis can be achieved by adding HCl as described in [41]. Additionally, NaOH was used to induce metabolic alkalosis which led to an increase in pH, as reported in [41].

### 3.3. Selection of ISF-Mimicking Solutions Based on Conductivity Measurements

After establishing the correlation between pH and impedance, the conductivity of different buffering solutions was calculated. Pockevicius et al. reported reference conductivity values of ISF, which range from 0.41–0.80 S/m [35]. Therefore, buffer solutions with a conductivity range of 0.41–0.80 S/m (BES, CBC, BTP, Trizma) can be used as ISF phantoms. As described in the following sections, firstly, the conductivity of the selected buffers was calculated by inducing acidosis and alkalosis using the methods proposed by Marin et al. The quantification of conductivity under different pH conditions allows the determination of the electrical behavior of the buffers, replicating the effects that would be expected in real ISF during acidosis and alkalosis. Secondly, to ensure the buffers are not only physiologically representative of real ISF (in terms of pH) under hypoxic conditions but also electrically representative, a comparison between the conductivity of the buffer solutions and the conductivity of real ISF was made. The comparison of the calculated conductivity value range to the reference conductivity values of ISF reported by Pockevicius et al. [35] allows the determination of which buffers are both electrically and physiologically representative of real ISF.

#### 3.3.1. Assessing Acidosis Effect through ISF-Mimicking Solutions with Actual pH Values

The impedance measurement results from the previous section showed that the impedance of the buffers decreases with an increase in pH and is highly correlated with the pH for CBC, BES, SPB, PPB, and PCB. For example, the BES buffer has a negative coefficient correlation (r) of −0.937 (*p*-value < 0.05). The pKa of the BES buffer is 7.09, and the useful pH range is 6.4–7.8. In this study, six solutions of BES buffer were prepared at pH values of 6.60, 6.75, 6.91, 7.05, 7.26, and 7.43. These buffers were prepared by changing the concentrations of their chemical reagents (BES and disodium hydrogen phosphate). Using the impedance measurements, and the cell constant of the probe, the conductivity for each buffer was calculated. The results shown in Table 3 suggest that the three solutions of BES buffer (6.75, 6.91, and 7.05) are within the conductivity range of ISF, i.e., 0.41–0.80 (S/m). Therefore, these three solutions of BES buffers were considered as ISF mimics. Additionally, an inverse relation was observed between the impedance and the pH value (impedance decreases as pH increases).

#### 3.3.2. Evaluating Acidosis Effect in pH-Adjusted Mimicked Solutions Using HCl

In the next set of experiments, a buffer with a pH range of > 7.30 was selected to introduce acidosis (asphyxia). This was achieved by adding HCl to decrease the pH of the buffer. The results have shown that among all the selected buffers (CBC, BES, SPB, PPB, and PCB), the CBC buffer was considered the suitable buffer to mimic the properties of ISF. The reason for the selection of the CBC buffer is due to its conductivity range being within the reference range of 0.41–0.80 (S/m), which corresponds to the pH range of 6–8. Eight CBC buffers with different pH values were prepared by changing the concentration of their chemical reagents (sodium bicarbonate and sodium carbonate anhydrous). The buffer with a suitable pH value of 9.47 (pH > 7.30) was selected for the introduction of HCl into the buffer. The impedance was measured after each addition of HCL. The results shown in Table 4 suggest that as a result of the decrease in pH from 7.27 to 6.36, the conductivity of the carbonate bicarbonate is decreased (0.80–0.69) and this remains within the reference values of ISF (pH is 6 to 8). After a decrease in pH (9.47 to 6.36), the impedance of the carbonate bicarbonate buffer is increased (33.61–45.61 Ω), which makes this buffer valid as an ISF-mimicking solution in the pH range 6.36–7.27 and conductivity range 0.69–0.80 S/m.

#### 3.3.3. Assessing Alkalosis Effect through ISF-Mimicking Solutions with Actual pH Values

In the following set of experiments, CBTP, BTP, and Trizma buffers were selected based on their pH and impedance relationship to assess the effect of alkalosis. These buffers were chosen without any adjustments to their pH levels. BTP buffer was prepared at 12 different pH levels (ranging from 6.01 to 8.43) by adjusting the concentrations of Bis-Tris propane and HCl. The impedance of each buffering solution was then measured to determine conductivity. The impedance of the BTP buffer increased from 38.32 to 54.62 Ω as pH increased from 6.01 to 8.43, as shown in Table 5. This makes the buffer suitable as an ISF-mimicking solution within the pH range of 6.41 to 7.81 and conductivity range of 0.77 to 0.60 S/m.

#### 3.3.4. Assessing Alkalosis Effect in pH-Adjusted Mimicked Solutions Using NaOH

In this set of experiments, a buffer was selected from CBTP, BTP, and Trizma to mimic the alkalosis effect in the ISF-mimicking solutions. The results demonstrated that BTP can successfully mimic the properties of ISF under alkalotic conditions, as presented in Table 6. For the experiments, a BTP buffer with a pH value of 6.23 was chosen, and NaOH was added while the pH and impedance values were measured. The BTP buffer was found to mimic the properties of ISF in the pH range of 6.23–7.73, with a reference conductivity range of 0.61–0.56 S/m and an increase in impedance with increasing pH. Furthermore, the BTP buffer is considered an ISF-mimicking solution within the pH range of 6.23–7.73.

Based on our experimental findings, it was found that the BES buffer and HCl-CBC buffer can mimic the metabolic acidosis effect of asphyxia, while the metabolic alkalosis effect can be mimicked by the BTP, Trizma, and NaOH-BTP buffers.

In biological tissues, alterations in pH can affect ionic conductivity by changing the concentration of ions responsible for electrical conductivity. Acidification, resulting in a decrease in pH, can increase the concentration of positively and negatively charged ions, leading to an increase in ionic conductivity. Conversely, alkalization, resulting in an increase in pH, can decrease the concentration of positively and negatively charged ions, leading to a decrease in ionic conductivity.

The three main factors that affect the conductivity of a solution are the concentrations of ions, the temperature of the solution, and the type of ions. Conductivity is also correlated with ionic mobility, which increases with the decrease in the size of the ion. For example, hydrogen ion (H+) has the highest conductivity and ionic mobility. So, the lower the pH, the higher the conductivity [42]. Our experiments validate that there is a relationship between the conductivity and pH of the ISF phantoms. The nature of this correlation (either positive or negative) depends on the characteristics of the buffers. The selection of buffers for mimicking ISF may also be influenced by the nature of the ions present in the chemical reagents. Additionally, the pH of a solution is dependent on its pKa value, and exceeding the buffering capacity may be another factor to consider when selecting an ISF mimic. The acidic or basic nature of the compound also impacts the pH of a solution, with a higher concentration of hydrogen ions resulting in a more acidic environment [41,42]. Boumann et al. reported that there is no significant difference in cerebral spinal fluid (CSF) conductivity across the frequency range of 10 Hz to 10 kHz at room temperature [29]. This observation is consistent with the similarities observed between ISF and CSF in terms of their composition, production, and fluid exchange mechanisms [30]. Consequently, our study’s findings align with those of [36,43] indicating that there is no significant difference in ISF conductivity across the frequency range of 10 Hz to 10 kHz at room temperature. Therefore, the electrical conductivity values of all the buffers are consistent.

## 4. Conclusions and Future Work

In this study, the potential of using ISF pH as a biomarker for hypoxia is considered. The basis for this consideration is two-fold: (1) pH levels quickly change in response to hypoxic events due to the limited buffering capacity of ISF; (2) there is potential to monitor ISF much less invasively compared to blood sampling. To develop novel sensors to monitor ISF pH, there is a requirement to develop electrically representative and pH-representative liquid mimics, and the evaluation of various candidate mimics is described.

In this study, eight buffers were prepared as potential ISF mimics, with pH values ranging from 6.00 to 8.00. All solutions were standardized to a concentration of 0.1 M, and experiments were conducted at room temperature. The candidate buffers were assessed in terms of their pH profile and their electrical conductivity compared to human ISF. Based on our experimental findings, it has been found that the BES buffer and HCl-CBC buffer can effectively mimic the metabolic acidosis induced by asphyxia. Conversely, the metabolic alkalosis effect can be mimicked using BTP, Trizma, and NaOH-BTP.

To the best of the authors’ knowledge, this is the very first study to design ISF-mimicking solutions based on physiological (pH) and electrical properties. As the fabrication of ISF phantoms to evaluate metabolic disorders (or asphyxia) or alkalosis based on electrical characterization is a relatively new approach, there are some limitations to our study. The study primarily focused on mimicking pH and electrical properties and did not directly consider other vital physiological parameters such as oxygen species and carbon dioxide levels. While the effect of these parameters is considered implicitly in terms of pH values, considering these additional parameters in preparing mimicking solutions would provide a more accurate physiological representation of ISF. Another limitation of this study is the lack of measurements of real human ISF for direct comparison of physiological and electrical properties of the interstitial-fluid-mimicking solutions and real human interstitial fluid samples. The practical constraints associated with obtaining sufficient quantities of human interstitial fluid present challenges in conducting these direct comparisons. This limitation restricts the ability to validate the developed interstitial-fluid-mimicking solutions against real ISF samples.

In summary, the current study provides valuable insights into the development of interstitial-fluid-mimicking solutions for hypoxia detection based on pH and electrical properties. However, to enhance the robustness and applicability, future studies can build upon this work by considering additional physiological parameters in preparation of mimicking solutions and incorporating direct comparisons between the prepared interstitial-fluid-mimicking solutions and actual human interstitial fluid samples. These additional analyses would strengthen the evaluation of the interstitial-fluid-mimicking solutions and provide further evidence of their suitability for pre-clinical assessments.

## Figures and Tables

**Figure 1 diagnostics-13-03125-f001:**
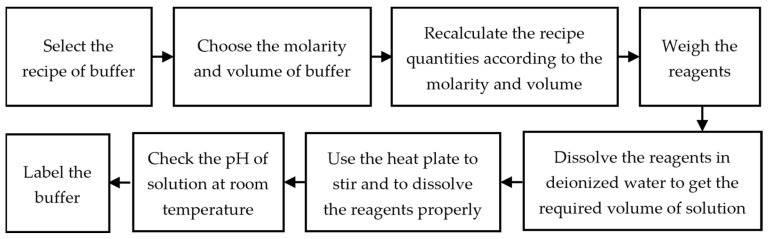
Flowchart of steps involved in the preparation of buffers.

**Figure 2 diagnostics-13-03125-f002:**
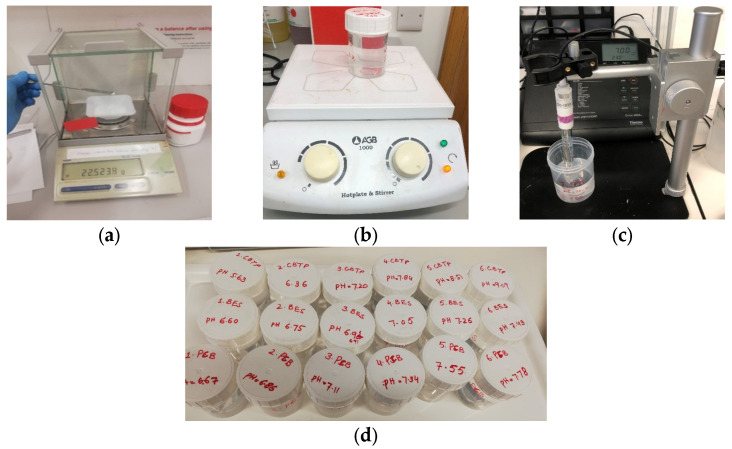
Preparation of buffers: (**a**) weighing the reagents, (**b**) stirring and dissolving the reagents, (**c**) recoding the pH at room temperature, (**d**) labeling the prepared buffers.

**Figure 3 diagnostics-13-03125-f003:**
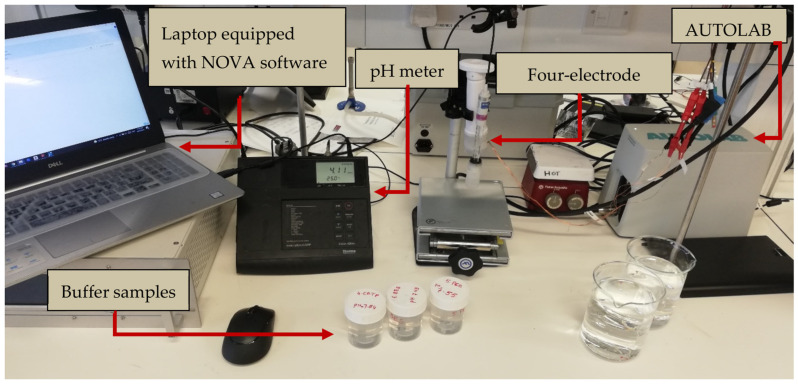
Impedance measurement setup: The measurement apparatus consists of a pH meter, a probe, several samples, and a laptop equipped with the NOVA software. In the background, the AUTOLAB is visible. To maintain the stability of the probe and cable, the samples are brought into proximity to the probe via the utilization of a lift table.

**Figure 4 diagnostics-13-03125-f004:**
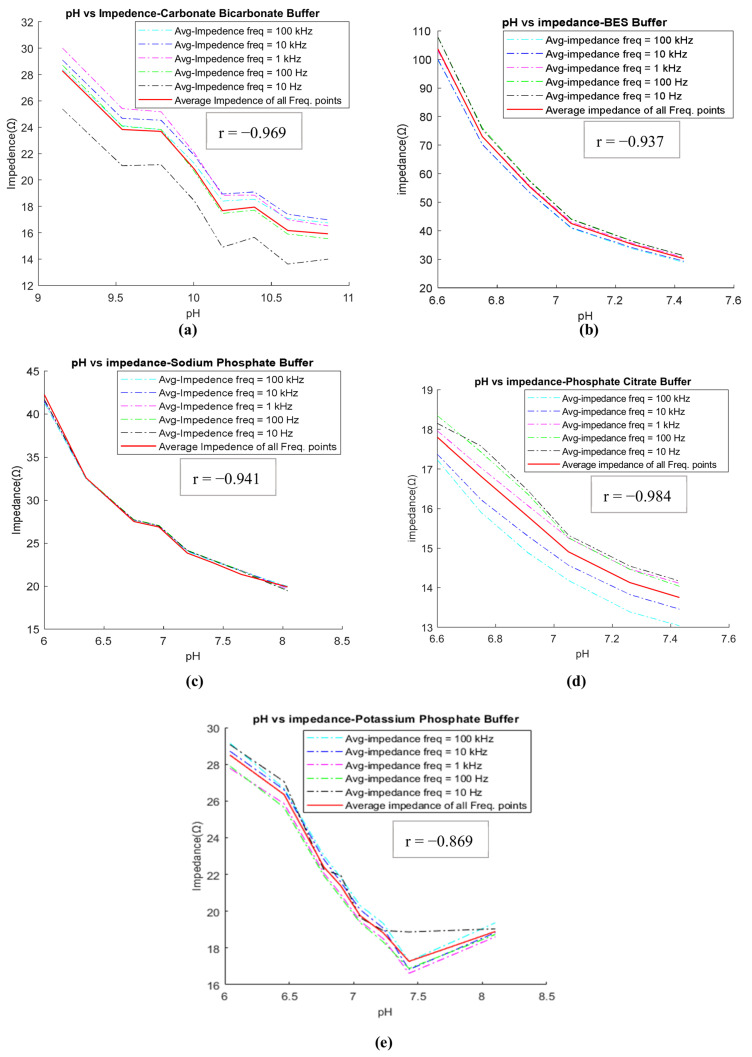
Impedance vs. pH relationship: impedance decreases with increasing pH in (**a**) CBC, (**b**) BES, (**c**) SPB, (**d**) PCB, and (**e**) PPB buffer solutions.

**Figure 5 diagnostics-13-03125-f005:**
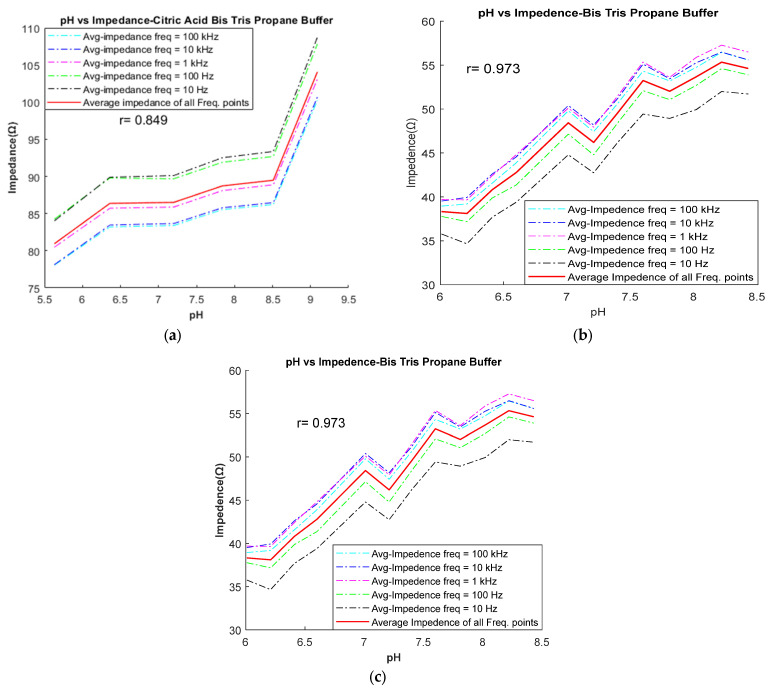
Impedance vs. pH relationship: impedance increases with increasing pH in (**a**) CBTP, (**b**) BTP, and (**c**) Trizma buffer.

**Table 1 diagnostics-13-03125-t001:** Buffers with their pH values.

S. No.	Buffer System	pH Range
1	Carbonate Bicarbonate Buffer (CBC)	9.2–10.6
2	BES	6.4–7.8
3	Sodium Phosphate (SPB)	5.8–7.4
4	Citrate Bis-Tris Propane (CBTP)	2.5–9.5
5	Potassium Phosphate (PPB)	5.8–8.0
6	Phosphate-Citrate (PCB)	2.2–8.0
7	Bis-Tris Propane (BTP)	5.8–7.2
8	Trizma	7.0–9.2

**Table 2 diagnostics-13-03125-t002:** Preparation of buffers (concentration: 0.1 M; volume: 0.1 L).

1. Carbonate Bicarbonate Buffer	
No. of buffers prepared: 8	
pH measured: 9.19, 9.54, 9.79, 10.01, 10.19, 10.39, 10.61, 10.87	
Reagents	Molecular Weight (g/mol)
Sodium Bicarbonate	84.01
Sodium carbonate (anhydrous)	105.99
2. BES	
No. of solutions prepared: 6	
pH measured: 6.60, 6.75, 6.91, 7.05, 7.26, 7.43	
Reagents	Molecular Weight (g/mol)
BES	213.3
Disodium hydrogen phosphate	141.96
3. Sodium Phosphate (SPB)	
No. of solutions prepared: 8	
pH measured: 6.00, 6.35,6.75, 6.96, 7.20, 7.41, 7.65, 8.04	
Reagents	Molecular Weight (g/mol)
Monosodium phosphate	141.96
Disodium phosphate	119.98
4. Citrate Bis-Tris Propane (CBTP)	
No. of solutions prepared: 6	
pH measured: 5.63, 6.36, 7.20, 7.84, 8.51, 9.09	
Reagents	Molecular Weight (g/mol)
Bis-Tris Propane	282.34
Citric acid	192.13
5. Potassium Phosphate (PPB)	
No. of solutions prepared: 8	
pH measured: 6.04, 6.46, 6.77, 6.90, 7.05, 7.23, 7.43, 8.10	
Reagents	Molecular Weight (g/mol)
Dipotassium phosphate	174.18
Potassium dihydrogen phosphate	136.086
6. Phosphate-Citrate (PCB)	
No. of solutions prepared: 6	
pH measured: 6.60, 6.75, 6.91, 7.05, 7.26,7.43	
Reagents	Molecular Weight (g/mol)
Disodium Hydrogen Phosphate	141.96
Citric Acid	210.14
7. Bis-Tris Propane (BTP)	
No. of solutions prepared: 12	
pH measured: 6.01, 6.21, 6.41, 6.60, 7.01, 7.21, 7.40, 7.60, 7.81, 8.02, 8.22,8.43	
Reagents	Molecular Weight (g/mol)
Bis-Tris Propane	282.33
HCl	36.46
8. Trizma	
No. of solutions prepared: 9	
pH measured: 6.77, 6.90, 7.23, 7.51, 7.73, 7.92, 8.13, 8.29, 8.49, 8.76, 9.22	
Reagents	Molecular Weight (g/mol)
Trizma HCl	157.6
Trizma Base	121.14
9. Artificial ISF [9]	
No. of solutions prepared: 1	
pH measured: 5.37	
Reagents	Molecular Weight (g/mol)
Calcium chloride	110.98
Glucose	180.156
HEPES	238.3012
Potassium chloride	74.5513
Magnesium sulfate	120.366
Sodium chloride	58.44
Sodium dihydrogen phosphate	119.98
Saccharose	342.3

**Table 3 diagnostics-13-03125-t003:** BES buffer mimicking acidosis effect.

pH	Conductivity (S/m)	Impedance (Ω)
6.60	0.30	103.69
6.75	0.43	72.96
6.91	0.56	55.44
7.05	0.73	42.60
7.26	0.89	35.08
7.43	1.03	30.30

**Table 4 diagnostics-13-03125-t004:** HCL_CBC buffer mimicking acidosis effect.

pH	Conductivity (S/m)	Impedance (Ω)
6.36	0.69	45.61
6.46	0.70	44.75
6.56	0.70	44.77
6.64	0.72	43.56
6.73	0.72	43.30
6.81	0.73	43.05
6.92	0.74	42.09
7.01	0.74	42.20
7.06	0.77	40.51
7.12	0.77	40.60
7.17	0.78	40.13
7.22	0.79	39.79
7.27	0.80	38.87
7.33	0.82	38.25
7.40	0.82	38.20
7.50	0.84	37.27
7.60	0.84	37.03
7.72	0.85	36.90
7.87	0.85	36.61
8.07	0.85	36.75
8.33	0.88	35.64
8.63	0.89	34.96
8.90	0.90	34.53
9.09	0.91	34.24
9.21	0.91	34.51
9.34	0.90	34.54
9.47	0.93	33.61

**Table 5 diagnostics-13-03125-t005:** BTP buffer mimicking alkalosis effect.

pH	Conductivity (S/m)	Impedance (Ω)
6.01	0.81	38.32
6.21	0.82	38.11
6.41	0.77	40.81
6.60	0.73	42.78
7.01	0.64	48.43
7.21	0.68	46.20
7.40	0.63	49.56
7.60	0.59	53.24
7.81	0.60	52.02
8.02	0.58	53.68
8.22	0.56	55.33
8.43	0.57	54.62

**Table 6 diagnostics-13-03125-t006:** NaOH-BTP buffer mimicking alkalosis effect.

pH	Conductivity (S/m)	Impedance (Ω)
6.23	0.61	50.96
6.53	0.61	51.39
6.76	0.60	52.14
6.96	0.59	53.25
7.06	0.58	53.66
7.27	0.57	54.69
7.47	0.57	54.55
7.73	0.56	55.56
8.07	0.55	56.93
8.43	0.55	56.72

## Data Availability

The article includes all the data as it has been both tabulated and plotted. Supplementary files can be provided for any additional data if necessary.

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
