# Peer review of "Development and Characterization of Interstitial-Fluid-Mimicking Solutions for Pre-Clinical Assessment of Hypoxia"

_diagnostics, 2023, doi:10.3390/diagnostics13193125_

Round 1
Reviewer 1 Report
Authors made artificial ISF for learning acidosis and alkalosis.
1. It is not convinced that impedance and conductivity test results could be used to find the best buffer. The model is too simple and there is no comparison between artifical buffer with real ISF.
2. The experiment of acidosis and alkalosis is not convincing either. A pH meter could be directly used rather than measuring impedance and conductivity.
English is fine.
Reviewer 2 Report
The authors are trying to develop interstitial fluid-mimicking solutions used for pre-clinical assessment of hypoxia, which is interesting. However, the experimental design in this work is so simple that it may not sufficiently support the conclusion. To enhance the soundness of this work, the reviewer strongly suggests that the author compare their interstitial fluid-mimicking solutions with actual human interstitial fluid samples, including the pH, impedance, oxygen species, and carbon dioxide. All these properties are vital parameters to determine if the prepared interstitial fluid can be efficiently used for the pre-clinical evaluation of hypoxia.
Round 2
Reviewer 2 Report
The authors addressed all the concerns the reviewer previously raised.